# Fresh and Aromatic Virgin Olive Oil Obtained from Arbequina, Koroneiki, and Arbosana Cultivars

**DOI:** 10.3390/molecules24193587

**Published:** 2019-10-05

**Authors:** Alfonso M. Vidal, Sonia Alcalá, Antonia De Torres, Manuel Moya, Juan M. Espínola, Francisco Espínola

**Affiliations:** Department of Chemical, Environmental and Materials Engineering, Universidad de Jaén, Campus Las Lagunillas, 23071 Jaén, Spain; amvidal@ujaen.es (A.M.V.); salcala@ujaen.es (S.A.); antorres@ujaen.es (A.D.T.); mmoya@ujaen.es (M.M.); juan1411994@hotmail.com (J.M.E.)

**Keywords:** super-high-density orchard, aroma, color, flavor, response surface methodology

## Abstract

Three factors for the extraction of extra virgin olive oil (EVOO) were evaluated: diameter of the grid holes of the hammer-crusher, malaxation temperature, and malaxation time. A Box–Behnken design was used to obtain a total of 289 olive oil samples. Twelve responses were analyzed and 204 mathematical models were obtained. Olives from super-intensive rainfed or irrigated crops of the Arbequina, Koroneiki, and Arbosana cultivars at different stages of ripening were used. Malaxation temperature was found to be the factor with the most influence on the total content of lipoxygenase pathway volatile compounds; as the temperature increased, the content of volatile compounds decreased. On the contrary, pigments increased when the malaxation temperature was increased. EVOO from irrigated crops and from the Arbequina cultivar had the highest content of volatile compounds. Olive samples with a lower ripening degree, from the Koroneiki cultivar and from rainfed crops, had the highest content of pigments.

## 1. Introduction

Spain, and particularly, the Spanish province of Jaén, is one of the great world powers in the production of virgin olive oil (VOO). Spanish olive oil production represents approximately 45% of the world total. Furthermore, due to its great extension, olive culture represents one of the principal economic sectors in Jaén. VOO is a fat that is known worldwide for its benefits to human health. Olive oil consumption in the Mediterranean diet is associated with the prevention of several diseases and low mortality from cardiovascular disease [1].

Furthermore, consumers greatly appreciate extra virgin olive oil (EVOO) for its aromatic characteristics and intense color. The specific flavor of EVOO defines its quality and uniqueness [2]. Consumers demand EVOO with a balanced composition. To achieve a balanced final product, several studies to optimize the extraction conditions and agronomic factors have been conducted [3,4,5].

The first aspects of olive oil that the consumer detects are its color and smell, which are important attributes in terms of the perception of food products. Chlorophylls and carotenoids are the main pigments responsible for the color of green olives and these photosynthetic pigments are present in olive oil [6]. Therefore, olive oil produced from green olives with a low degree of maturation will take on a green color, while more mature purple olives impart a golden color. The volatile compounds present in VOO play an important role in the perception of the odor of VOO. The volatile C6 and C5 aldehydes, alcohols, and esters in VOO are mainly produced via the lipoxygenase (LOX) pathway; the formation of these compounds begins due to cell disruption during the crushing of the olives and continues throughout the extraction process [7,8]. More than 180 different aromatic compounds have been identified in olive oils [9] but only a small fraction of the large number of volatile compounds present in olive oil actually contribute to its overall aroma. In addition, these compounds are responsible for the positive attributes of olive oils and are indicators of a high-quality EVOO [10]. Phenolic compounds play an important role in the intensity of the release of certain aroma compounds during the consumption of EVOO. These compounds interact with certain volatile compounds; the resulting complexes probably reduce their volatility during the organoleptic perception of olive oil [11].

During olive oil production, many factors influence the extraction process and product quality. The cultivar of olives used is one of the most important factors and significantly influences the volatile compound composition. Growing the same olive cultivar in different locations results in oils with different volatile profiles [9]. In addition to the cultivar, the irrigation management, stage of maturity, and conditions used in the extraction process are other principal factors. The malaxation time and temperature are among the main technological variables during the extraction process. Modifying the EVOO extraction conditions can change the oil composition [12]. Agronomic practices are also a relevant factor. Traditionally, olive trees have been grown under rainfed conditions. However, new crops are adapted to irrigation techniques, increasing the total production of VOO [13].

The objective of this work was to determine the optimum extraction conditions in terms of maximizing the content of volatile compounds and pigments in the oil using a response surface methodological approach. The obtained mathematical models allow mathematical optimization, producing virgin olive oil that is desirable to consumers. Mathematical models were developed for agronomic factors, namely, the use of irrigation or rain-feeding and the ripening time of the olives, for the Arbequina, Koroneiki, and Arbosana cultivars.

## 2. Results and Discussion

Table 1 shows the results of the characterization of the samples. The samples had maturity index (MI) values ranging from 0 to 3. As expected, the oil content increased with an increasing MI of the olive samples. Samples from rainfed crops produced a higher percentage of olive oil than irrigated ones, probably due to the higher proportion of moisture present in the irrigated samples. On the other hand, irregular rainfall that occurred during the olive crop harvest due to climate change was probably the reason that the moisture and solids content did not follow a linear trend.

Considering the number of olive samples in Table 1 and the runs of each sample, a total of 289 oils were obtained and 12 responses, including volatile compounds and photosynthetic pigments, were analyzed for each oil. The tables and figures presented in this paper are only an example of the results obtained.

As an example, the experimental values corresponding to a sample of the Arbequina cultivar (irrigated and MI = 1.11) obtained from the twelve responses studied for each olive sample are shown in Table 2. Seventeen similar tables were produced in this work. These results were processed using the Design-Expert software and mathematical models were obtained. These mathematical models can help to predict the optimal values of the technological factors to maximize each response within the range of factors studied. Table 3 shows the proposed models in terms of the actual factors with the statistical parameters and optimal values of the technological factors to obtain maximum response values for a sample of the Arbequina cultivar (irrigated, MI = 1.11). Seventeen tables similar to Table 3 were produced in this work.

Table 4 shows the maximum values for each of the volatile compound and pigment responses for all samples. Finally, Table 5 shows the optimal values of the technological factors related to the extraction for two important responses: the total volatile compound and chlorophyll contents.

### 2.1. LOX Pathway Volatile Compounds

Among the agronomic factors studied, the stage of maturation had only a small effect on the total volatile compounds, probably because the maturation stages studied were very similar (Table 4). However, the differences between the oils obtained from irrigated and rainfed olives were significant, as were those among the oils derived from different cultivars. Similar differences were reported by Gómez-Rico et al. [14] in a study involving the Arbequina, Cornicabra, Morisca, Picolimon, Picudo, and Picual cultivars. Olive oils originating from irrigated crops had a higher content of volatile compounds than those from rainfed crops for a given cultivar. Comparing the three cultivars, the irrigated Arbequina (MI = 1.11) sample had the highest content of volatile compounds at 34.94 mg/kg; the sample with the lowest content was the rainfed Koroneiki sample (MI = 1.68) at 6.77 mg/kg, which was five times smaller. According to the results obtained by García et al. [15], the oils from irrigated orchards had higher contents of (*E*)-2-hexenal, the major volatile compound, than oils from rainfed orchards.

According to Fregapane et al. [16], the volatile compounds most affected by irrigation are (*E*)-2-hexenal, (*Z*)-3-hexenol, and hexanol. The increase in the amount of water supplied to the olive trees significantly increases the contents of these volatile compounds. It is noteworthy that the (*E*)-2-hexenal content of the Arbequina cultivar samples was very high (29.39 mg/kg for the irrigated sample with MI = 1.11), compared with that of the Koroneiki samples (4.37 mg/kg for the irrigated sample with MI = 0.67); these compounds are very important contributors to the delicate green perception of EVOO. Our results are similar to those obtained by Cherfaoui et al. [17], which showed that the contents of volatile compounds and (*E*)-2-hexenal increased with maturity, reaching a maximum concentration when the color of the olive fruit skin changed from purple to black. As the MI increased, the contents of linoleic and linolenic FA decreased, whereas the contents of volatile compounds from the LOX pathway increased. This could be attributed to the consumption of these two FA, which are used as substrates for the LOX enzyme present in the olive pulp [18].

In terms of the technological factors, the malaxation temperature had the greatest influence on the total content of LOX pathway volatile compounds; as the temperature increased, the volatile compound contents decreased. Thus, low temperatures are required to obtain the highest content of volatile compounds (Figure 1, Figure 2 and Figure 3). This enzymatic pool is sensitive to its environmental conditions; the temperature, in particular, can affect the level and the activity of the enzymes involved in the LOX pathway [19]. According to the mathematical models obtained, 20 °C (the lower limit of the range studied) was the optimum working temperature for most of the samples (Table 5). However, Ridolfi et al. [20] studied the kinetic constants of the olive LOX enzyme and reported that the maximum LOX activity was recorded at 30 °C. Also, in agreement with the results of Angerosa et al. [21] and Kalua et al. [9], we observed that increased malaxing temperature increased the content of hexanol and (*E*)-2-hexenol (Table 3), which is considered to impart a highly unpleasant odor.

Different authors have reported widely different conclusions regarding the effect of malaxing time on the volatile compound content, with some saying that the malaxing time should be short, while others recommend increasing malaxing time. The malaxing time influences the activity of the enzymes and the vaporized compounds in the ambient atmosphere [22]. The recommendations for malaxing time differ because the interactions with malaxing temperature have not been sufficiently studied. At low temperatures, the content of volatile compounds increases with increasing time, while at high temperatures it decreases, as shown in Figure 1 and Figure 3. Thus, to obtain the highest content of volatile compounds at low temperatures, a long malaxing time of 90 min was required (upper limit of the range studied), as shown for most of the samples in Table 5. However, the malaxing time was not a significant factor in four mathematical models.

Just as the malaxing time had little influence on the mathematical models, as seen in the perturbation graphs of Figure 1, Figure 2 and Figure 3, the diameter of the crusher holes did not appear in eight of the models. In the models in which it did appear, the maximum volatile compound content was obtained using the upper limit size, 6.5 mm, in some cases and the smallest size, 4.5 mm, in others. This would seem to indicate that the diameter did not have a significant influence on the volatile content of the olive oils.

According to Cevik et al. [10] to obtain the maximum quantity of (*E*)-2-hexenal, a lower temperature and long malaxation time should be used. We do not agree with the results obtained by Lukić et al. [5], which indicated that (*E*)-2-hexenal and 1-penten-3-one content increased with higher malaxation temperatures.

### 2.2. Photosynthetic Pigments

Table 4 shows the maximum values of the chlorophyll and carotenoid content predicted by the models. In addition to being responsible for the color of olive oil, these natural pigments also play an important role in the oxidative stability of the oil [23]. The sample with the highest content of chlorophylls was the Koroneiki rainfed sample and that with the lowest content was the Arbequina irrigated sample. The sample with the highest content of carotenoids was the Koroneiki rainfed sample, while the Arbequina sample had the lowest carotenoid content. García et al. [15] reported that EVOO produced from rainfed crops exhibited a higher level of photosynthetic pigments than EVOO from irrigated crops. Samples with lower MIs had a higher content of these photosynthetic compounds. During the ripening process, the chlorophyll and carotenoid contents decreased, which was similar to the results observed by Benito et al. [24]. To obtain olive oils with a green color, i.e., with a high chlorophyll content, green or less mature olives should be used in the extraction. In addition, the rainfed samples contained more chlorophylls than the irrigated ones. These results also coincided with those obtained by Benito et al. [24,25].

The decrease in the chlorophyll and carotenoid content during the ripening process is due to the involvement of an enzymatic system in the degradation of chlorophylls during the maturation of olive fruits, as described by Vergara-Domínguez et al. [26]. The ripeness stage of the olives at harvesting was correlated to the amount of pigments in the resulting EVOOs. Interestingly, a decrease in the ratio of chlorophyll derivatives to carotenoids in the olive oils was observed with increasing maturity of the olives at harvesting [25].

In terms of the technological factors, in general, the adjustment of the mathematical models for both photosynthetic pigments was good and the three factors studied had a great influence on the contents of chlorophylls and carotenoids, as can be seen in the perturbation graph in Figure 4. Essentially, to obtain a high content of photosynthetic compounds, a malaxation temperature of 40 °C, a malaxation time of 90 min, and a hammer-crusher grid hole diameter of 4.5 mm should be used. The same conditions were obtained for the Arbequina, Koroneiki, and Arbosana cultivars. The temperature and malaxation time had a great influence; when both were increased, the chlorophyll content of the resulting oil increased. On the contrary, the diameter had less influence, with increasing diameter decreasing pigment content.

Our models are different from those obtained by Brahim et al. [27] using the same methodology (RSM); their model did not indicate an influence of the temperature on the chlorophyll content, probably due to the fact that they used a different cultivar, Chemlali, at a different maturity index, 4.7.

## 3. Materials and Methods

### 3.1. Raw Material

Olive fruits (*Olea europaea* L.) were picked by hand from super-intensive crops located in El Carpio (Córdoba, Spain) between October and December 2015. In total, 17 samples of olives were collected. Each sample was composed of approximately 10 kg of olives. The olives were harvested at different ripening degrees from olive trees of the Arbequina, Koroneiki, and Arbosana cultivars that had been either irrigated or rainfed. The maturity index (MI) or ripening degree was determined according to the method of Uceda and Frías, as described in Espínola et al. [28]. The olives were harvested with maturity indices between 0 and 3 to prevent the fruits from falling onto the ground. The oil content was analyzed by the Soxhlet method and the moisture content was determined by drying at 105 °C. All measurements were carried out quadruplicate; the results are presented in Table 1.

### 3.2. Olive Oil Extraction

Olive oils were obtained using an Abencor centrifugal system (Abencor analyser, MC2 Ingeniería y Sistemas S.L., Seville, Spain) under laboratory-scale conditions, as previously described by Espínola et al. [29]. The obtained olive oils were decanted in a graduated tube for at least three hours, filtered using paper, and stored in amber glass bottles at −18 °C under a nitrogen atmosphere until their analyses.

### 3.3. Analysis of Photosynthetic Pigments: Chlorophylls and Carotenoids

The composition of the pigments was determined following the procedure proposed by Minguez-Mosquera et al. [30]. The absorbance was measured using a Shimadzu UV-1800 spectrophotometer (Shimadzu Corp., Kyoto, Japan); a wavelength of 470 nm was used for the carotenoid pigments and 670 nm for the chlorophyllic pigments. Cyclohexane was used as the solvent. Equation (1) has been used to obtain the pigments concentration.
(1)$$C_{p}=\frac{A_{\lambda }{\cdot V}_{f}}{{\varepsilon \cdot \;m}_{a}}\;\times 10000$$
where C_p_ is the concentration of the pigment (mg of pigment / kg of oil); A_λ_, is the absorbance at 670 nm for chlorophyll and 470 nm for carotenoids; ε is the specific absorbance, for chlorophyll is 613 and for carotenoids is 2000; m_a_ is the weight of the sample (g) and V_f_ is the volume of the solution. The pigment concentration of the samples is expressed as mg of pigment per kg of oil.

### 3.4. Analysis of Volatile Compounds

The procedure for the determination of the volatile compounds was similar to that used by Vidal et al. [31]. The volatile compounds were analyzed by headspace solid-phase microextraction (HS-SPME) and gas chromatography-flame ionization detection (GC-FID). Two grams of the sample was placed in a 20 mL amber glass vial tightly capped with a polytetrafluoroethylene (PTFE)/silicone septum and a magnetic cap. The vial was heated to 40 °C for 10 min to allow the volatile compounds to reach equilibrium in the headspace. Subsequently, the SPME needle was inserted through the septum and the fiber was exposed for 40 min. The SPME fiber (2 cm length and 50/30 μm film thickness) was purchased from Supelco (Bellefonte, PA, USA) and was composed of Carboxen/DVB/polydimethylsiloxane (PDMS). The fiber had been previously conditioned following the instructions of the manufacturer.

GC-FID analysis was performed using a gas chromatograph (model 7890B, Agilent Technologies, CA, USA) equipped with a split/splitless injector and a flame ionization detector. The volatile compounds adsorbed on the fiber were desorbed in the injector port for 1 min in splitless mode. A DB-WAXetr polyethylene glycol capillary column (30 m length, 0.25 mm internal diameter, 0.25 μm coating) (Agilent Technologies, USA) was used for the chromatographic separation. The carrier gas was helium at a flow rate of 1 mL/min. The injector temperature was 260 °C and that of the detector was 280 °C. The oven temperature was initially set to 40 °C and was held at this temperature for 10 min. Subsequently, the temperature was increased to 160 °C with a temperature ramp of 3 °C/min, followed by a ramp of 15 °C/min to 200 °C; the sample was then held at 200 °C for 5 min.

The chromatographic peaks were quantified using the internal standard method. 4-Methyl-2-pentanol was used as the internal standard, and all the compounds were used as external standards. The internal standard solution was prepared in previously deodorized oil. Then, the standard was added to each oil sample and stirred so that the mixture is homogeneous. The results are expressed as mg of standard compound per kg of oil.

### 3.5. Experimental Design and Statistical Analysis

The statistical design of experiments (SDE) and response surface methodology (RSM) statistical tools were used for planning the methodology and data analysis. The optimal experimental strategy was used to obtain the most information with the minimum cost. RSM facilitates the evaluation of results and generates reliable conclusions [32]. Different experimental designs can be used with RSM; in this case, a Box−Behnken design was used for the optimization of three factors: the diameter of the grid holes of the hammer-crusher, the malaxation temperature, and the malaxation time, with 17 runs including five central points. The extraction conditions were a crusher hole diameter of 4.5, 5.5, or 6.5 mm; a malaxation temperature of 20, 30, or 40 °C; and a malaxation time of 30, 60, or 90 min. Table 6 shows the values of the factors.

The response variables studied were: 9 individual volatile compounds, the sum of the compounds (TOTAL), and 2 pigments (chlorophylls and carotenoids).

The experimental results were analyzed using the software Design-Expert v. 8.0.7.1 (Stat-Ease, Inc., Minneapolis, MN, USA). The coefficient of determination (R^2^), the lack of fit, and the Fisher value (F-value) were obtained from the analysis of variance (ANOVA) and were used to determine the adequacy of the proposed model. Equation (2), a quadratic model, was used for each response studied.
Y = β_0_ + β_1_ D + β_2_ T + β_3_ t + β_12_ D T + β_13_ D t + β_23_ T t + β_11_ D^2^ + β_22_ T^2^ + β_33_ t^2^ ± SD(2)
where D is the grid hole diameter of the crusher (mm), T is the malaxation temperature (°C), and t is the malaxation time (min). Y is the predicted response and was correlated with the set of coefficients (β): the intercept (β_0_), linear (β_1_, β_2_, β_3_), interaction (β_12_, β_13_, β_23_), and quadratic (β_11_, β_22_, β_33_). The *p*-value was established as 5% (*p*-value ≤ 0.05). SD is the standard deviation.

## 4. Conclusions

The results of this study suggest that the samples harvested from irrigated crops had the highest total content of LOX pathway volatile compounds. Comparing the three cultivars analyzed, the Arbequina cultivar had the highest content of these compounds. Samples harvested from rainfed crops had a higher content of photosynthetic pigments than irrigated samples. The Koroneiki cultivar had the highest content of chlorophylls and carotenoids. Therefore, it is not possible to optimize the irrigation management strategy to obtain an olive oil that is both rich in volatile compounds and high in photosynthetic pigments. Olive samples with a lower ripening degree had the highest content of these photosynthetic compounds, as well as of the total LOX pathway volatile compounds. The maturation stage had only a small influence, probably because the maturation stages studied were very similar.

In relation to the optimal conditions of each cultivar, it has been determined that they are practically the same for the three cultivars. So for maximum volatile compound content, we should work with irrigated olives at the lowest malaxation temperature, 20 °C, malaxation time, 90 min, and a small grid hole diameter, 4.5 mm, for a higher chlorophyll content, the same conditions except the temperature that must be the highest malaxation temperature, 40 °C.

The malaxation temperature was the factor that had the greatest influence on the total LOX pathway volatile compounds. Lower malaxation temperature resulted in a higher quantity of volatile compounds. On the other hand, the chlorophyll content increased with increasing temperature and malaxation time. Therefore, it is not possible to optimize the malaxing temperature to obtain an olive oil that is both rich in volatile compounds and high in photosynthetic pigments.

## Figures and Tables

**Figure 1 molecules-24-03587-f001:**
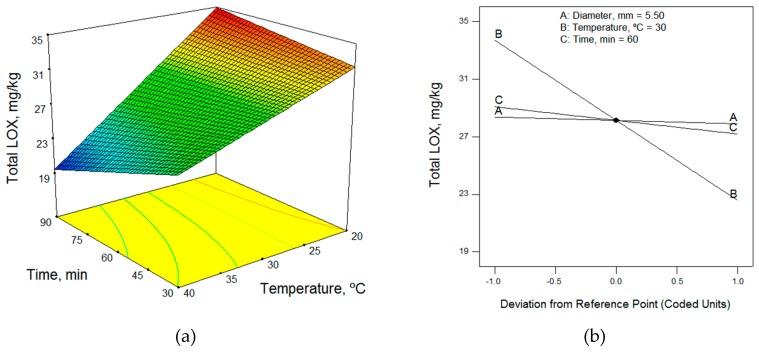
(**a**) Response surface and (**b**) perturbation graphic of the effects of temperature and time on the total LOX volatile content for the Arbequina cultivar (irrigated, MI = 1.11), for a diameter of 5.5 mm.

**Figure 2 molecules-24-03587-f002:**
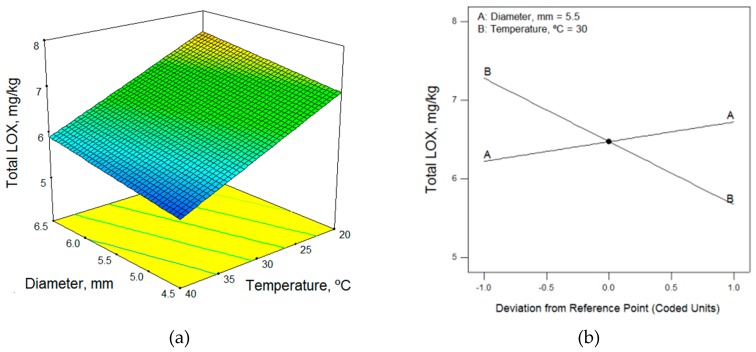
(**a**) Response surface and (**b**) perturbation graphic of the effects of temperature and time on the total LOX volatile content for the Koroneiki cultivar (rainfed, MI = 2.05); time does not have an influence.

**Figure 3 molecules-24-03587-f003:**
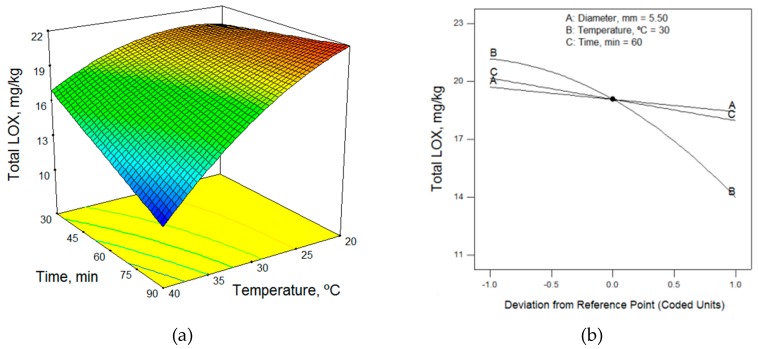
(**a**) Response surface and (**b**) perturbation graphic of the effects of temperature and time on the total LOX volatile content for the Arbosana cultivar (irrigated, MI = 0.07), for a diameter of 5.5 mm.

**Figure 4 molecules-24-03587-f004:**
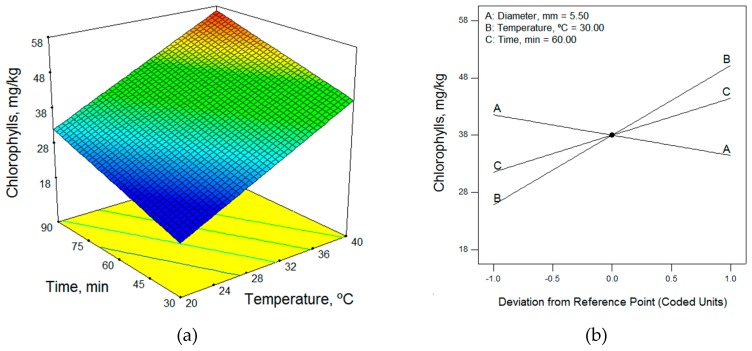
(**a**) Response surface and (**b**) perturbation graphic of the effects of temperature and time on the chlorophyll content for the Koroneiki cultivar (rainfed, MI = 0.16), for a diameter of 5.5 mm.

**Table 1 molecules-24-03587-t001:** Compositional characteristics and maturity index of the processed olives *.

Variety	Irrigation	Maturity Index (MI)	Moisture (%)	Oil (%)	Solids (%)
Arbequina	Rainfed	0.20	53.72 ± 1.13 ^a^	15.89 ± 0.49 ^a^	30.39 ± 0.68 ^a^
		1.31	55.88 ± 0.38 ^b^	17.78 ± 0.10 ^b^	26.34 ± 0.39 ^b^
		2.56	54.96 ± 0.78 ^b^	20.05 ± 0.59 ^c^	24.99 ± 0.81 ^c^
	Irrigated	0.16	60.93 ± 0.70 ^c^	12.36 ± 0.25 ^d^	26.71 ± 0.63 ^b,d^
		1.11	61.78 ± 0.43 ^c,d^	15.13 ± 0.19 ^e^	23.08 ± 0.47 ^e^
		2.52	61.96 ± 0.28 ^d,e^	16.19 ± 0.28 ^a^	21.85 ± 0.23 ^f^
Koroneiki	Rainfed	0.16	53.11 ± 0.85 ^a,f^	16.06 ± 0.18 ^a^	30.82 ± 0.81 ^a^
		1.68	53.45 ± 0.56 ^a,f^	21.73 ± 0.15 ^f^	24.82 ± 0.59 ^c^
		2.05	52.60 ± 0.59 ^f^	20.41 ± 0.19 ^c^	26.99 ± 0.62 ^b,d^
	Irrigated	0.07	59.91 ± 0.78 ^g^	11.65 ± 0.29 ^g^	28.44 ± 0.83 ^g^
		0.67	59.03 ± 0.40 ^g^	16.25 ± 0.17 ^a^	24.71 ± 0.53 ^c^
		2.30	53.29 ± 0.56 ^a,f^	19.38 ± 0.34 ^h^	27.33 ± 0.87 ^d^
Arbosana	Rainfed	0.15	55.38 ± 0.87 ^b^	17.23 ± 0.38 ^i^	27.39 ± 0.65 ^d^
		0.95	59.89 ± 0.64 ^g^	15.92 ± 0.38 ^a^	24.19 ± 0.54 ^c,h^
		2.11	54.99 ± 0.41 ^b^	21.64 ± 0.17 ^f^	23.38 ± 0.54 ^e,h^
	Irrigated	0.07	63.19 ± 0.32 ^h^	12.67 ± 0.10 ^d^	24.14 ± 0.38 ^c,h^
		0.58	62.8 ± 0.54 ^e,h^	14.53 ± 0.3 ^j^	22.67 ± 0.58 ^e,f^
Fisher’s LSD			0.94	0.42	0.88

* Means of five replicates ± SD. The letters represent different groups. There were no statistically significant differences among those groups that share the same letter. The method used to discriminate between the means was Fisher’s least significant difference (LSD).

**Table 2 molecules-24-03587-t002:** Experimental results for the irrigated Arbequina cultivar (MI = 1.11).

	Lipoxygenase (LOX) Pathway Volatile Compounds (mg/kg)										Pigments (mg/kg)	
Run	*(E)*-2-Hexenal	Hexanal	*(Z)*-3-Hexenol	Hexanol	*(E)*-2-Hexenol	*(Z)*-2-Pentenol	1-Penten-3-ol	1-Penten-3-ona	*(Z)*-3-Hexenyl acetate	TOTAL	Chlorophylls	Carotenoids
1	23.27	1.24	0.73	0.34	0.34	0.52	0.35	0.55	0.52	27.87	5.07	4.38
2	28.48	2.10	0.77	0.49	0.32	0.56	0.43	0.62	0.43	34.19	4.60	4.61
3	23.89	1.82	0.58	0.41	0.38	0.64	0.45	0.62	0.60	29.39	5.88	4.80
4	19.82	1.75	1.10	0.78	0.63	0.58	0.37	0.52	0.47	26.03	5.02	4.09
5	15.81	0.97	1.09	0.51	0.67	0.53	0.33	0.47	0.33	20.70	7.22	4.58
6	26.49	1.84	0.74	0.61	0.21	0.53	0.38	0.69	0.44	31.93	2.83	3.36
7	29.16	2.05	1.19	0.30	0.40	0.54	0.38	0.64	0.29	34.96	3.36	3.55
8	14.58	0.94	0.77	0.62	0.53	0.53	0.33	0.45	0.47	19.22	4.02	5.01
9	19.36	0.80	0.55	0.33	0.37	0.52	0.35	0.50	0.32	23.10	8.28	5.53
10	21.14	0.83	0.78	0.46	0.36	0.55	0.39	0.57	0.48	25.55	5.39	4.45
11	25.23	0.96	1.09	0.49	0.41	0.51	0.33	0.58	0.31	29.92	3.76	3.44
12	24.77	0.96	0.52	0.46	0.21	0.56	0.40	0.66	0.46	29.01	5.50	4.86
13	24.88	1.28	0.71	0.43	0.34	0.53	0.36	0.55	0.50	29.58	4.20	4.00
14	26.23	1.93	0.65	0.56	0.21	0.61	0.49	0.73	0.57	31.98	4.18	4.26
15	23.86	0.91	0.72	0.35	0.35	0.52	0.35	0.54	0.50	28.11	4.94	4.08
16	24.13	1.27	0.71	0.32	0.37	0.55	0.37	0.58	0.45	28.76	5.90	4.59
17	23.35	1.14	0.66	0.43	0.36	0.54	0.35	0.56	0.51	27.90	6.36	4.86

**Table 3 molecules-24-03587-t003:** Proposed models in terms of actual factors with statistical parameters (analysis of variance (ANOVA) for the fit of experimental data was used) and optimal values of the technological factors for the response maximum values. Irrigated Arbequina cultivar sample (MI = 1.11).

Response	Model *	*p*-Value	R^2^	Std. Dev.	Maximum Value (mg/kg)	Diameter (mm)	Temperature (°C)	Time (min)
LOX pathway volatile compounds								
*(E)*-2-Hexenal (mg/kg)	−8.407 + 6.601 D + 0.8259 T + 0.3760 T − 0.1622 D T − 0.03773 D t − 7.124E−03 T t	<0.0001	0.967	0.91	29.38	6.46	20.01	41.34
Hexanal (mg/kg)	+2.48071 -0.054697 T +8.33412E-003 t	<0.0001	0.841	0.20	2.11	In range	20.26	88.51
*(Z)*-3-Hexenol (mg/kg)	+3.080 − 0.9373 D −0.03426 T + 0.1099 D^2^ + 5.392E−04 T^2^	<0.0001	0.982	0.03	1.16	6.50	20.00	In range
Hexanol (mg/kg)	+3.843 − 0.4551 D − 0.07405 T − 0.04065 t + 0.01079 D T + 2.845E−03 D t + 2.366E−04 T t + 1.589E−04 T^2^	0.0101	0.813	0.07	0.81	6.41	37.65	89.94
*(E)*-2-Hexenol (mg/kg)	+1.5301 − 0.5926 D − 8.738E−03 T + 1.022E−03 t + 2.729E−03 D T + 5.891E−05 T t + 0.0571 D^2^	<0.0001	0.987	0.02	0.71	6.49	37.62	86.61
*(Z)*-2-Pentenol (mg/kg)	+0.5495	-	--	0.03	0.55	In range	In range	In range
1-Penten-3-ol (mg/kg)	+1.978 − 0.4505 D − 0.01744 T + 8.377E−04 t + 2.275E−03 D T + 0.03150 D^2^	<0.0001	0.988	0.01	0.52	4.50	20.00	90.00
1-Penten-3-one (mg/kg)	+1.125 − 0.03819 D − 8.740E−03 T − 1.235E−03 t	<0.0001	0.895	0.03	0.73	4.61	20.34	31.62
*(Z)*-3-Hexenyl acetate (mg/kg)	+0.45091	-	--	0.09	0.45	In range	In range	In range
TOTAL (mg/kg)	+1.171 + 5.934 D + 0.6129 T + 0.3793 t − 0.1344 D T − 0.03562 D t − 7.165E−03 T t	<0.0001	0.978	0.81	34.94	6.50	20.00	90.00
*Pigments*								
Chlorophylls (mg/kg)	+1.825 − 0.5600 D + 0.1709 T + 0.02394 t	0.0002	0.803	0.70	8.30	4.50	40.00	90.00
Carotenoids (mg/kg)	+4.960 − 0.4732 D + 0.04738 T + 0.01001 t	0.0001	0.792	0.29	5.54	4.50	39.91	81.92

* According to coefficients of model (Equation (1)), only significant values are included (*p*-value < 0.005); D is the grid hole diameter of the crusher (mm), T is the malaxation temperature (°C), and t is the malaxation time (min); R^2^ is the determination coefficient, Std. Dev. is standard deviation.

**Table 4 molecules-24-03587-t004:** Maximum value for volatile compounds and pigments response *.

			LOX Pathway Volatile Compounds (mg/kg)										Pigments (mg/kg)	
Cultivar	Irrigation	MaturityIndex (MI)	(*E*)-2-Hexenal	Hexanal	(*Z*)-3-Hexenol	Hexanol	(*E*)-2-Hexenol	(*Z*)-2-Pentenol	1-Penten-3-ol	1-Penten-3-one	(*Z*)-3-Hexenyl acetate	TOTAL	Chlorophylls	Carote noids
Arbequina	Rainfed	0.20	9.66 ± 0.85	4.24 ± 0.58	0.87 ± 0.04	0.33 ± 0.04	1.18 ± 0.06	0.79 ± 0.05	0.76 ± 0.04	0.71 ± 0.04	0.73 ± 0.09	17.29 ± 1.66	32.24 ± 2.06	13.61 ± 0.66
		1.31	13.93 ± 0.47	1.56 ± 0.10	0.87 ± 0.03	0.52 ± 0.16	0.63 ± 0.04	0.64 ± 0.01	0.54 ± 0.02	0.71 ± 0.03	0.51 ± 0.06	18.48 ± 0.58	11.23 ± 0.54	6.91 ± 0.29
		2.56	13.23 ± 0.36	1.53 ± 0.08	0.65 ± 0.05	0.49 ± 0.14	0.64 ± 0.01	0.52 ± 0.03	0.47 ± 0.03	0.70 ± 0.03	0.48 ± 0.07	17.90 ± 0.68	13.65 ± 1.15	5.52 ± 0.30
	Irrigated	0.16	18.53 ± 1.09	6.87 ± 0.10	1.48 ± 0.08	0.31 ± 0.07	0.59 ± 0.04	0.62 ± 0.08	0.52 ± 0.11	0.53 ± 0.07	0.53 ± 0.08	27.83 ± 0.98	18.42 ± 0.63	9.14 ± 0.38
		1.11	29.39 ± 0.91	2.10 ± 0.20	1.16 ± 0.03	0.80 ± 0.07	0.67 ± 0.02	0.55 ± 0.03	0.52 ± 0.01	0.74 ± 0.03	0.45 ± 0.09	34.94 ± 0.81	8.30 ± 0.70	5.56 ± 0.29
		2.52	26.52 ± 0.76	2.05 ± 0.05	0.87 ± 0.02	0.36 ± 0.03	0.67 ± 0.03	0.51 ± 0.01	0.39 ± 0.01	0.58 ± 0.02	0.74 ± 0.09	31.80 ± 1.20	18.27 ± 0.63	9.06 ± 0.39
Koroneiki	Rainfed	0.16	2.85 ± 0.09	2.55 ± 0.40	0.89 ± 0.03	0.27 ± 0.06	0.63 ± 0.02	0.78 ± 0.04	0.64 ± 0.05	0.64 ± 0.04	1.15 ± 0.08	10.03 ± 0.52	60.13 ± 4.06	23.50 ± 1.34
		1.68	1.84 ± 0.09	0.47 ± 0.04	0.57 ± 0.03	0.79 ± 0.05	0.57 ± 0.03	0.85 ± 0.02	0.59 ± 0.08	0.90 ± 0.05	0.59 ± 0.01	6.77 ± 0.16	50.06 ± 0.91	15.95 ± 0.76
		2.05	2.17 ± 0.11	0.54 ± 0.01	0.67 ± 0.03	0.54 ± 0.18	0.48 ± 0.07	0.93 ± 0.02	0.90 ± 0.01	0.97 ± 0.03	0.68 ± 0.02	7.53 ± 0.32	49.20 ± 1.61	18.27 ± 1.62
	Irrigated	0.07	4.22 ± 0.46	3.96 ± 0.61	1.01 ± 0.07	0.47 ± 0.02	0.82 ± 0.03	0.53 ± 0.10	0.47 ± 0.17	0.40 ± 0.07	0.61 ± 0.13	10.66 ± 1.35	42.83 ± 3.59	18.03 ± 1.11
		0.67	4.37 ± 0.24	0.62 ± 0.17	1.17 ± 0.03	0.43 ± 0.07	0.79 ± 0.03	0.62 ± 0.04	0.66 ± 0.03	0.70 ± 0.03	0.49 ± 0.04	9.71 ± 0.40	46.15 ± 1.58	14.38 ± 1.03
		2.30	5.44 ± 0.24	0.79 ± 0.04	0.83 ± 0.04	0.46 ± 0.08	0.68 ± 0.01	0.74 ± 0.05	0.72 ± 0.04	0.90 ± 0.04	0.91 ± 0.11	10.24 ± 0.64	54.38 ± 1.94	18.15 ± 0.61
Arbosana	Rainfed	0.15	11.77 ± 0.29	0.99 ± 0.09	0.68 ± 0.02	0.47 ± 0.05	0.33 ± 0.11	0.63 ± 0.01	0.53 ± 0.02	0.72 ± 0.02	1.00 ± 0.07	16.40 ± 0.33	12.08 ± 0.82	6.96 ± 0.29
		0.95	10.29 ± 0.34	0.56 ± 0.08	0.63 ± 0.02	0.47 ± 0.10	0.53 ± 0.02	0.55 ± 0.01	0.49 ± 0.03	0.76 ± 0.05	0.41 ± 0.07	14.49 ± 0.60	10.97 ± 0.83	6.65 ± 0.34
		2.11	11.21 ± 0.29	0.92 ± 0.04	0.69 ± 0.08	0.44 ± 0.09	0.48 ± 0.12	0.56 ± 0.06	0.41 ± 0.03	0.87 ± 0.05	1.84 ± 0.24	17.11 ± 0.31	10.43 ± 0.55	6.60 ± 0.32
	Irrigated	0.07	17.45 ± 0.62	2.19 ± 0.06	1.14 ± 0.05	0.50 ± 0.14	0.62 ± 0.01	0.57 ± 0.03	0.48 ± 0.02	0.64 ± 0.02	0.48 ± 0.08	22.62 ± 0.72	12.45 ± 0.85	7.10 ± 0.35
		0.58	16.86 ± 0.54	0.80 ± 0.25	1.17 ± 0.05	0.44 ± 0.12	0.62 ± 0.02	0.57 ± 0.01	0.43 ± 0.03	0.65 ± 0.02	0.49 ± 0.06	22.73 ± 0.43	11.04 ± 0.81	6.63 ± 0.28

* Value ± Standard deviation of the model.

**Table 5 molecules-24-03587-t005:** Optimal values for volatile compound and chlorophylls response.

			Total LOX Pathway Volatile Compounds				Chlorophylls			
Cultivar	Irrigation	Maturity Index (MI)	Maximum Value * (mg/kg)	Diameter (mm)	Temperature (°C)	Time (min)	Maximum Value * (mg/kg)	Diameter (mm)	Temperature (°C)	Time (min)
Arbequina	Rainfed	0.20	17.29 ± 1.66	In range	20.00	In range	32.24 ± 2.06	4.50	40.00	90.00
		1.31	18.48 ± 0.58	In range	25.91	50.47	11.23 ± 0.54	4.50	40.00	90.00
		2.56	17.90 ± 0.68	4.50	25.24	90.00	13.65 ± 1.15	4.50	40.00	90.00
	Irrigated	0.16	27.83 ± 0.98	In range	20.00	90.00	18.42 ± 0.63	4.50	40.00	90.00
		1.11	34.94 ± 0.81	6.50	20.00	90.00	8.30 ± 0.70	4.50	40.00	90.00
		2.52	31.80 ± 1.20	In range	20.00	90.00	18.27 ± 0.63	4.50	40.00	90.00
Koroneiki	Rainfed	0.16	10.03 ± 0.52	6.50	20.00	90.00	60.13 ± 4.06	4.50	40.00	90.00
		1.68	6.77 ± 0.16	6.50	20.00	90.00	50.06 ± 0.91	4.50	40.00	90.00
		2.05	7.53 ± 0.32	6.50	20.00	In range	49.20 ± 1.61	4.50	40.00	88.56
	Irrigated	0.07	10.66 ± 1.35	In range	20.00	In range	42.83 ± 3.59	4.50	40.00	90.00
		0.67	9.71 ± 0.40	In range	20.00	30.00	46.15 ± 1.58	In range	40.00	90.00
		2.30	10.24 ± 0.64	In range	21.53	In range	54.38 ± 1.94	4.50	40.00	90.00
Arbosana	Rainfed	0.15	16.40 ± 0.33	In range	22.36	49.77	12.08 ± 0.82	4.50	40.00	90.00
		0.95	14.49 ± 0.60	4.50	20.00	55.97	10.97 ± 0.83	4.50	40.00	89.99
		2.11	17.11 ± 0.31	4.50	26.41	90.00	10.43 ± 0.55	4.50	40.00	90.00
	Irrigated	0.07	22.62 ± 0.72	4.50	20.00	90.00	12.45 ± 0.85	4.50	40.00	90.00
		0.58	22.73 ± 0.43	4.50	20.00	90.00	11.04 ± 0.81	4.50	40.00	90.00

* Value ± Standard Deviation of the model.

**Table 6 molecules-24-03587-t006:** Actual factors of the experimental design (coded factors).

Run	Diameter (mm)	Temperature (°C)	Time (min)
1	5.5 (0)	30 (0)	60 (0)
2	5.5 (0)	20 (−1)	90 (+1)
3	4.5 (-1)	30 (0)	90 (+1)
4	6.5 (+1)	30 (0)	90 (+1)
5	6.5 (+1)	40 (+1)	60 (0)
6	5.5 (0)	20 (−1)	30 (−1)
7	6.5 (+1)	20 (−1)	60 (0)
8	5.5 (0)	40 (+1)	90 (+1)
9	4.5 (−1)	40 (+1)	60 (0)
10	5.5 (0)	40 (+1)	30 (−1)
11	6.5 (+1)	30 (0)	30 (−1)
12	4.5 (−1)	30 (0)	30 (−1)
13	5.5 (0)	30 (0)	60 (0)
14	4.5 (−1)	20 (−1)	60 (0)
15	5.5 (0)	30 (0)	60 (0)
16	5.5 (0)	30 (0)	60 (0)
17	5.5 (0)	30 (0)	60 (0)

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
