# Peer review of "Fresh and Aromatic Virgin Olive Oil Obtained from Arbequina, Koroneiki, and Arbosana Cultivars"

_molecules, 2019, doi:10.3390/molecules24193587_

Round 1

Reviewer 1 Report

The manuscript applies the Response Surface Methodology approach in order to determine the optimum extraction conditions for maximizing the content of volatile compounds and pigments in extra virgin olive oil and produce a virgin olive oil that is desirable to consumers. The study is very interesting and appealing to the consumers; the research design is appropriate and the results are well presented. My recommendation is accept after minor revision.

Here below, you will find some minor corrections:

Abstract

L.11 ....of extra virgin olive oil (EVOO)....

Results and Discussion

L.70: The samples had Maturity Index (MI)....

L.151: Our results are similar with those....

L.207: Delete ...that....

Reviewer 2 Report

The article show a lot of data about the chemical characterization of several olive samples. The authors should improve the results discussion, in addition lots of question about the statistical analysis must be answered. 

Materials and Methods 

Provide more information in Section 3.3, i.e. details about the pigments analysis Section 3.4.: how was used the internal standard? Add this into the SPME vial is difficult, and their evaporation could saturate the vapor phase. Please, supply more details about this process. Section 3.5.: what were the response variables? The abstract cited 12 responses, how those were selected?

Results 

Table 1: what samples were statistics different? Show in the Table  Table 2: why M. I. 1.11?  Line 87: where are these tables? Supplementary files?  All responses are significatives? How was stablished that?  Why performed a RSA for each variable? Why did not a multiresponse analysis?  Table 5: optimal values according what response? This is not clear in the manuscript  Figures did not show optimal responses   

Author Response

Please see the attchment

Reviewer 3 Report

Comments and Suggestions for Authors:

In this paper, the factors for the extraction of extra virgin olive oil were optimized, including diameter of the grid holes of the hammer-crusher, malaxation temperature, and malaxation time. The authors have designed a number of parameters and variables to find out the factors that influence the volatile compounds and photosynthetic pigments of extra virgin olive oil. However, I still have some questions that need to be clarified.

In conclusion, authors suggested that “it is not possible to optimize the irrigation management strategy to obtain an olive oil that is both rich in volatile compounds and high in photosynthetic pigments” and “it is not possible to optimize the malaxing temperature to obtain an olive oil that is both rich in volatile compounds and high in photosynthetic pigments.” The authors can discuss whether it is better to retain volatile compounds or photosynthetic pigments in high-quality extra virgin olive oil. The final optimal extraction conditions for virgin olive oil obtained from the Arbequina, Koroneiki and Arbosana cultivars can be summarized in conclusion. The determination of the volatile compounds were analyzed by gas chromatography-flame ionization detection (GC-FID). In table 4, nine volatile compounds were detected. Is there any unknown volatile compound? If the unknown substance is not included, the total amount of volatile compounds may be incorrect.

Round 2

Reviewer 2 Report

The corrections are suitable, I consider that the the article can be published.